# CALL TO ECLS—Acronym for Reporting Patients for Extracorporeal Cardiopulmonary Resuscitation Procedure from Prehospital Setting to Destination Centers

**DOI:** 10.3390/healthcare12161613

**Published:** 2024-08-13

**Authors:** Tomasz Sanak, Mateusz Putowski, Marek Dąbrowski, Anna Kwinta, Katarzyna Zawisza, Andrzej Morajda, Mateusz Puślecki

**Affiliations:** 1Faculty of Health Sciences, Jagiellonian University Medical College, 31-008 Cracow, Poland; 2Department of Anesthesiology and Intensive Care, University Hospital in Cracow, 30-688 Cracow, Poland; 3Collegium Medicum, Jan Kochanowski University, 25-317 Kielce, Poland; 4Department of Medical Education, Poznan University of Medical Sciences, 60-806 Poznan, Poland; 5Department of Anesthesiology and Intensive Care, Jagiellonian University Medical College, 31-501 Cracow, Poland; 6Epidemiology and Preventive Medicine, Jagiellonian University Medical College, 31-034 Cracow, Poland; 7Department of Medical Rescue, Poznan University of Medical Sciences, 60-608 Poznan, Poland; 8Department of Cardiac Surgery and Transplantology, Poznan University of Medical Sciences, 61-848 Poznan, Poland

**Keywords:** emergency medicine, pre-hospital care, communications, checklists in emergency medicine, cardiac arrest, extracorporeal cardiopulmonary resuscitation, eCPR, cardiopulmonary resuscitation

## Abstract

The acronym CALL TO ECLS has been proposed as a potential tool to support decision-making in critical communication moments when qualifying a patient for the ECPR procedure. The aim of this study is to assess the accuracy of the acronym and validate its content. Validation is crucial to ensure that the acronym is theoretically correct and includes the necessary information that must be conveyed by EMS during the qualification of a patient with out-of-hospital cardiac arrest for ECMO. A survey was conducted using the LimeSurvey platform through the Survey Research System of the Jagiellonian University Medical College over a 6-month period (from December 2022 to May 2023). Usefulness, importance, clarity, and unambiguity were rated on a 4-point Likert scale, from 1 (not useful, not important, unclear, ambiguous) to 4 (useful, important, clear, unambiguous). On the 4-point scale, the Content Validity Index (I-CVI) was calculated as the percentage of subject matter experts who rated the criterion as having a level of importance/clarity/validity/uniqueness of 3 or 4. The Scale-level Content Validity Index (S-CVI) based on the average method was computed as the average of I-CVI scores (S-CVI-AVE) for all considered criteria (protocol). The number of fully completed surveys by experts was 35, and partial completion was obtained in 63 cases. All criteria were deemed significant/useful, with I-CVI coefficients ranging from 0.87 to 0.97. Similarly, the importance of all criteria was confirmed, as all I-CVI coefficients were greater than 0.78 (ranging from 0.83 to 0.97). The average I-CVI score for the ten considered criteria in terms of usefulness/significance and importance exceeded 0.9, indicating high validity of the tool/protocol/acronym. Based on the survey results and analysis of responses provided by experts, a second version was created, incorporating additional explanations. In Criterion 10, an explanation was added—“Signs of life”—during conventional cardiopulmonary resuscitation (ROSC, motor response during CPR). It has been shown that the acronym CALL TO ECLS, according to experts, is accurate and contains the necessary content, and can serve as a system to facilitate communication between the pre-hospital environment and specialized units responsible for qualifying patients for the ECPR.

## 1. Introduction

The incidence of out-of-hospital cardiac arrest (OHCA) in Europe ranges from 350,000 to 700,000 events annually, which gives an average of 55 to 113 events per 100,000 residents. Unfortunately, the overall survival rate reaches only 2% to 11% and shows regional variability [1]. Positive outcomes can be achieved in cases of sudden cardiac arrest (SCA) with an initial shockable rhythm, followed by an early defibrillation and high-quality cardiopulmonary resuscitation (CPR) aimed at minimizing interruptions in chest compression and addressing potentially reversible causes of cardiac arrest. In such cases, return of spontaneous circulation (ROSC) can occur in as many as 30% to 40% of cases [2]. For patients with refractory cardiac arrest, extracorporeal cardiopulmonary resuscitation (ECPR) may be considered. European Resuscitation Council (ERC) guidelines reflect a growing body of evidence supporting extracorporeal cardiopulmonary resuscitation as a beneficial rescue therapy for selected cardiac arrest patients when conventional advanced life support (ALS) is ineffective, and specific interventions are available within a short timeframe, such as coronary angiography and percutaneous coronary intervention, pulmonary thrombectomy for massive pulmonary embolism, or rewarming in hypothermic cardiac arrest [3,4]. CPR with extracorporeal perfusion improves blood flow and oxygenation of vital organs, as well as supporting the central nervous system, preventing irreversible organ damage and cerebral ischemia during cardiac arrest. Inclusion criteria for published studies on ECPR vary, but most often include age ranges 18–65 or 18–75, cardiac arrest in the presence of witnesses, immediate bystander CPR, initial shockable rhythm, and estimated time from the start of resuscitation to ECPR initiation (low-flow time) <60 min. A shorter low-flow time consistently correlates with improved survival [5]. Appropriate application of ECPR in OHCA increases the chances of patient survival with favorable neurological outcomes [6]. The timing of team activation and cannulation performance plays an extremely important role in this context. One of the issues that can contribute to the prolongation of the time is inadequate communication between the emergency medical team and the destination center.

Effective communication within a multidisciplinary team is crucial for enhancing patient safety and the quality of work in emergency situations [7,8]. Research has shown that communication failures may come from several reasons, including difficulties among junior team members in opening up to senior specialists due to concerns about incompetence, embarrassment, verbal reprimand, lack of prior preparation or personal non-technical skills (NTSs). Poor communication is noticeable across various healthcare settings, with a significant focus on the moment of patient handover or transfer to different departments, when it is essential to share key health information in a short timeframe [9]. Such situations often occur in the Emergency Department as well [10]. To improve and systematize communication, communication strategies containing easy-to-remember acronyms are desired. These acronyms allow for the systematic inclusion of crucial patient-related information for further therapeutic processes in a short amount of time. One of these acronyms is SBAR (situation, background, assessment, and recommendation) and AT-MIST, along with the medical history acronym SAMPLE [11,12,13]. In the context of this article, the pivotal point is the handover of the patient from the emergency medical team to the Emergency Department or a telephone consultation with a specialist.

Efforts should be made to shorten the patient handover time while simultaneously conveying the most pertinent health information, allowing the medical staff to adequately prepare for the patient’s arrival. Our devised acronym CALL TO ECLS has been proposed as a potential tool to support decision-making in critical moments to facilitate the communication process. The validation of the acronym was conducted to ensure that it includes all necessary elements according to expert opinions to quickly and effectively qualify a patient for ECPR. Our study provides the first evidence that the proposed acronym is theoretically sound and can be implemented by emergency medical teams as a communication tool. The details of the acronym are provided in Table 1.

## 2. Objective of the Study

The objective of this study is to assess the CALL TO ECLS acronym and validate its content to ensure that the acronym encompasses an appropriate number of elements and adequately covers the scope of the studied field.

## 3. Materials and Methodology

The survey was designed using the LimeSurvey platform through the Survey Research System of the Jagiellonian University Medical College and was conducted over a 6-month period (from December 2022 to May 2023). The questionnaire was composed of two parts: I. Metrics (7 questions) and II. Assessment of the significance/relevance of individual elements of the CALL TO ECLS acronym (4 questions). Detailed questions are presented in Table 2. Participants were allowed to skip questions. An email containing a survey link was sent to experts specializing in Extracorporeal Life Support (ECLS). Experts were intentionally recruited from corresponding authors on PubMed, among those who have area of expertise in ECLS, as well as members of the European branch of the Extracorporeal Life Support Organization (EuroELSO).

To obtain expert evaluations of the proposed criteria, a structured validity assessment form was prepared and sent to them. This form aimed to assess the usefulness and importance of each criterion in relation to the purpose of the devised acronym, as well as the clarity and unambiguity of their phrasing. Usefulness, importance, clarity, and unambiguity were rated on a 4-point Likert scale ranging from 1 (not useful, not important, unclear, and ambiguous) to 4 (useful, important, clear, and unambiguous). Experts were also asked to propose additional criteria or suggest the removal of any of the existing ones.

## 4. Statistical Methods

On the 4-point scale, the Content Validity Index (I-CVI) was calculated as the percentage of subject matter experts who rated the criterion as having a level of importance/clarity/validity/uniqueness of 3 or 4. The Scale-level Content Validity Index (S-CVI) based on the average method was computed as the average of I-CVI scores (S-CVI-AVE) for all considered criteria (protocol). Additionally, the percentage of criteria that all experts rated as important (3 or 4) was calculated using the Universal Agreement among Experts (S-CVI-UA). Any I-CVI value of 0.5 or less should be excluded from the tool, as this value is unacceptable. For at least nine experts, an acceptable I-CVI is at least 0.78. An S-CVI-AVE (and S-CVI-UA) of at least 0.8 indicates an accepted value, while an S-CVI-AVE (and S-CVI-UA) of at least 0.90 indicates excellent content validity of the tool. Nevertheless, S-CVI-UA is sensitive to the number of experts, and with an increase in the number of experts, it is more likely to generate a low score. It is recommended that both indicators are presented. These are common criteria used in content validity assessments [14]. These statistical methods were used to evaluate the content validity of the CALL TO ECLS acronym and its individual elements based on expert assessments and their agreement on the importance, clarity, and relevance of the proposed criteria.

## 5. Results

The preliminary version of the tool was sent to 287 experts. The final number of fully completed surveys was 35, and partial completion was obtained in 63 cases. The average professional experience among the surveyed experts was 15 years (median: 12 years). Specialist doctors comprised 86% of the respondents, with 67% being specialists in anesthesiology and intensive care, 17% in emergency medicine, and 16% in cardiology/cardiac surgery. The average annual number of ECPR qualifications performed by surveyed doctors was 5.7 (median: 4). The respondents primarily came from Europe (69%, n = 68), with the majority from Poland (49%, n = 48), followed by the Czech Republic (10%, n = 10), Germany (7%, n = 7), Belgium (2%, n = 2), and Sweden (1%, n = 1). Two experts were from China and one from Japan. Twenty-seven experts did not specify their workplace.

A total of 67 respondents answered Question 7, which concerned the preparedness of the person reporting the patient’s condition for ECPR qualification. Among them, 43% (n = 29) indicated that the reporting person was completely unprepared for reporting the patient’s condition. Detailed results are presented in Figure 1.

These findings illustrate the demographic characteristics of the surveyed experts, their experience levels, and their responses to specific questions related to the assessment of the CALL TO ECLS acronym.

All criteria were deemed significant/useful, with I-CVI coefficients ranging from 0.87 to 0.97. Similarly, the importance of all criteria was confirmed, as all I-CVI coefficients were greater than 0.78 (ranging from 0.83 to 0.97). Regarding clarity assessment, four out of ten criteria had I-CVI values slightly below 0.78. The lowest I-CVI value of 0.74 was recorded for Criterion 3, pertaining to life-saving interventions, and Criterion 10, related to “signs of life”. This suggests that these criteria raised some doubts among experts regarding the clarity of the conveyed content; however, the value is far from 0.5, which would indicate the need for content reformulation. For Criteria 8 (context of the situation) and 9 (life-limiting diseases), the I-CVI values were 0.78. The lowest value for the unambiguity indicator (0.66) was observed for Criterion 10, similar to clarity assessment. Criteria 8, 9, and 2 were also rated slightly below 0.78 in terms of unambiguity (with I-CVI values of 0.71, 0.78, and 0.77, respectively).

The average I-CVI score for the ten considered criteria in terms of usefulness/significance and importance exceeded 0.9, indicating high validity of the tool/protocol/acronym. Similarly, the S-CVI-AVE values for clarity and unambiguity were higher than the required 0.80, at 0.85 and 0.81, respectively. All S-CVI-UA results were significantly below 0.8 (ranging from 0.31 for uniqueness to 0.63 for significance). Detailed results are presented in Table 3.

In response to Question 2 in Part II (Do you suggest including any additional point in the CALL TO ECLS report?), experts provided several significant suggestions for adding the following elements (n = 35): temperature (61%), pH value (30%), and potential poisoning (9%).

## 6. Modifications

Based on the survey results and analysis of responses provided by experts, a second version was created, incorporating additional explanations. In Criterion 10, an explanation was added—“Signs of life”—during conventional CPR (ROSC, motor response during CPR). Due to feedback suggesting the addition of elements to the acronym, relevant clarifications were sent. The entire algorithm refers to normothermic patients, and the lack of critical parameter analyzers in most pre-hospital medical rescue systems means that the pH parameter cannot be assessed in pre-hospital care at this time. Adding this parameter to the acronym would compromise its universality. The authors decided to include the central body temperature value; however, ongoing efforts are being made to create a dedicated acronym for reporting patients in deep hypothermia with concurrent circulatory–respiratory instability or cardiac arrest. Version 2.0 of the acronym is presented in Table 4.

## 7. Discussion

An important correlation between age, initial VF/pVT rhythm, and the likelihood of survival and good neurological outcomes has been established [15,16,17]. Survival after ECPR is also influenced by the time between CA and the initiation of ECMO support, referred to as low-flow time, during which organ perfusion is impaired, potentially leading to multi-organ failure and cerebral ischemia [18,19,20]. The application of ECPR improves blood flow and oxygen delivery to tissues, preventing irreversible organ damage, especially in cases of central nervous system ischemia during CA [21]. Favorable ECPR outcomes are observed when extracorporeal techniques are applied rapidly (ideally within 1 h of CA) and when the arrest is due to potentially reversible causes. Two models of out-of-hospital ECPR implementation have been adopted worldwide [17,22]. The “Prague model” involves ECMO implantation in a hospital setting, while the “Paris model” involves ECMO implantation at the scene of the event before transporting the patient to a hospital for further intervention. The benefits of ECPR are evident when timely interventions, such as percutaneous coronary intervention in acute coronary syndrome or pulmonary thrombectomy in massive pulmonary embolism, are performed [23,24]. Correct ECPR application in refractory OHCA increases survival chances to 25–35%, with most cases achieving good neurological outcomes [25].

Extracorporeal circulation is also employed for patients after in-hospital cardiac arrest (IHCA). In these cases, factors such as time to identify CA, personnel and equipment availability significantly influence the procedure’s effectiveness. IHCA patients differ significantly from OHCA patients; they are predominantly older and have multiple comorbidities. The initial shockable rhythm is less common in IHCA compared to OHCA (38% vs. 59%). Signs of deteriorating vital parameters may appear hours before CA, highlighting the importance of rapid response teams, known as Medical Emergency Teams (METs). Despite the challenges associated with IHCA patients, the use of ECPR is justified. Improved survival rates and good neurological outcomes at 3 months post-arrest indicate the benefits of ECPR over conventional resuscitation techniques (21% vs. 11%). Moreover, maintaining ECMO after successful resuscitation can offer additional advantages [26,27,28,29].

Communication errors can be prevented, enhancing patient safety in high-risk healthcare settings [30]. Acronyms serve as useful guides to prevent communication errors across various medical fields [31]. Around 60–70% of high-risk situation errors result from “human factors” [32]. CALL TO ECLS was developed to reduce communication errors and streamline the qualification process for ECPR by pre-hospital medical teams. Medical personnel can be trained to use the clear information provided in our acronym. The use of standardized tools can balance communication styles among different healthcare professionals and minimize critical intervention response times, such as ECPR [33,34,35].

Regarding the first version of the protocol, I-CVI and S-CVI-AVE values for importance and relevance, coupled with strong conceptual and developmental frameworks, indicate the high content validity of the criteria and protocol. However, I-CVI values for clarity and unambiguity suggest some criteria revision. The S-CVI-AVE indices for clarity and unambiguity demonstrate acceptable content validity of the protocol. While the S-CVI-UA values did not meet acceptability criteria, this coefficient is sensitive to expert numbers. Given the study’s sample size of 35 experts, S-CVI-UA should be treated as additional information. In this case, S-CVI-AVE is recommended. The use of the CVI index is preferred due to its advantages over other indices. However, one weakness of the CVI index is its lack of correction for chance agreement; thus, modified Kappa statistics are recommended. Nevertheless, in this study, due to a large expert sample [34], the chance agreement was close to zero and therefore the modified Kappa was very close to the reported I-CVI.

## 8. Limitations

The study presents an evaluation of the CALL TO ECLS acronym and is merely a proposal for a quick communication tool. It includes the necessary information about the patient that must be conveyed to qualify the patient for the ECPR procedure. However, qualification criteria may vary between countries, which may necessitate modifications to the acronym. Another limitation of the study is the small number of survey responses, caused by the still limited prevalence of this procedure both in Poland and Europe.

## 9. Conclusions

It has been shown that the CALL TO ECLS acronym, according to experts, is accurate and contains the necessary content, and can serve as a system to facilitate communication between the pre-hospital environment and specialized units responsible for qualifying patients for the ECPR procedure. Our acronym will not only assist in practical patient handovers in prehospital settings, but will also be useful for the academic community during medical simulations. It includes all the necessary elements that students will need to convey during simulations to qualify a patient for ECPR.

## Figures and Tables

**Figure 1 healthcare-12-01613-f001:**
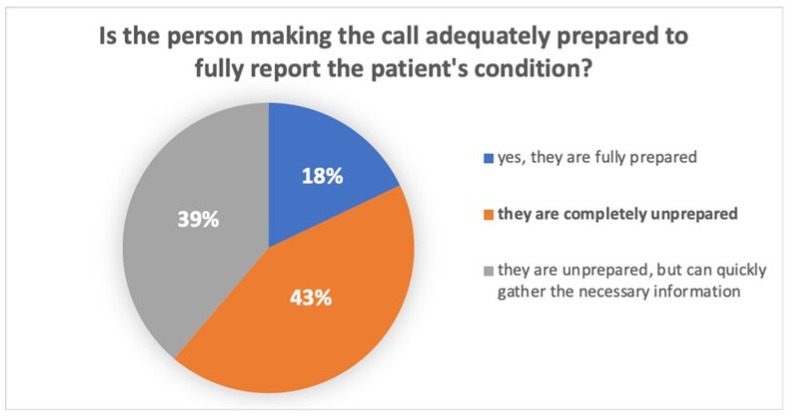
Is the calling person prepared to be able to fully report the patient’s condition?

**Table 1 healthcare-12-01613-t001:** Detailed description of the CALL TO ECLS acronym Version 1.0.

Acronym CALL TO ECLS	
**C**	**C**ardiac Arrest to first **C**PR (“no-flow interval”) < 5 min
**A**	**A**ge, Physical **A**ctivity
**L**	**L**ife-saving interventions Performed
**L**	**L**evel of end-tidal CO_2_ during CPR
**T**	**T**rauma signs
**O**	**O**btained rhythm
**E**	**E**stimated time of arrival to the hospital
**C**	**C**ontext of the situation
**L**	**L**ife-**L**imiting comorbidities
**S**	“**S**igns of life” during conventional CPR

**Table 2 healthcare-12-01613-t002:** Survey questions.

Question	I. Metrics
1.	Please specify the country in which you work.
2.	Please provide your workplace.
3.	Please provide the level of referral of the hospital in which you work.
4.	You work as (please select from the list): a specialist in anaesthesiology and intensive care physician resident of anaesthesiology and intensive care physician of another specialty (please specify)
5.	Please provide your seniority in years
6.	What is the average annual number of qualifications for ECPR performed by you
7.	Is the calling person prepared to be able to fully report the patient’s condition?
	**II. Assessment of the Significance/Relevance of Individual Elements of the CALL TO ECLS Report**
1.	Please rate the individual indicators on a scale from 1 to 4: 4 means = useful/important/clear/unambiguous 3 means = rather useful/rather important/rather clear/rather unambiguous 2 means = rather not useful/rather not important/rather not clear/rather not unambiguous 1 means = not useful/not important/not clear/not unambiguous
2.	Do you suggest including any additional point in the CALL TO ECLS report?
3.	Do you suggest deleting any existing item out of the CALL TO ECLS report?
4.	What additional parameters do you propose to add to the acronym CALL TO ECLS if the patient is reported from a Hospital Emergency Department/Trauma Center/other hospital unit?

**Table 3 healthcare-12-01613-t003:** Content validity of each criterion and the entire protocol in terms of usefulness/significance, importance, clarity, and unambiguity.

No.	Criterion	Usefulness		Importance		Clarity		Unambiguity	
		No. of Ratings 3 or 4	I-CVI	No. of Ratings 3 or 4	I-CVI	No. of Ratings 3 or 4	I-CVI	No. of Ratings 3 or 4	I-CVI
1	**C**ardiac arrest to first **C**PR (“no-flow interval”) < 5 min	34	0.97	34	0.97	34	0.97	32	0.91
2	**A**ge, Physical **A**ctivity	33	0.94	30	0.86	30	0.86	27	0.77
3	**L**ife-saving interventions performed	34	0.97	33	0.94	26	0.74	29	0.83
4	**L**evel of end-tidal CO_2_ during CPR	33	0.94	31	0.89	34	0.97	33	0.94
5	**T**rauma signs	32	0.91	30	0.86	30	0.86	28	0.80
6	**O**btained rhythm	32	0.91	32	0.91	31	0.87	32	0.91
7	**E**stimated time of arrival to the hospital	32	0.91	33	0.94	32	0.91	28	0.80
8	**C**ontext of the situation	31	0.87	32	0.91	27	0.77	25	0.71
9	**L**ife-**L**imiting comorbidities	33	0.94	32	0.91	27	0.77	26	0.74
10	“**S**igns of life” during conventional CPR	32	0.91	29	0.83	26	0.74	23	0.66
	S-CVI-AVE		0.93		0.90		0.85		0.81
	S-CVI-UA		0.63		0.57		0.51		0.31

**Table 4 healthcare-12-01613-t004:** Detailed description of the CALL TO ECLS acronym Version 2.0.

Acronym CALL TO ECLS	
**C**	**C**ardiac arrest to first **C**PR (“no-flow interval”) < 5 min
**A**	**A**ge, Physical **A**ctivity
**L**	**L**ife-saving interventions performed
**L**	**L**evel of end-tidal CO_2_ during CPR, level of central temperature
**T**	**T**rauma signs
**O**	**O**btained rhythm
**E**	**E**stimated time of arrival to the hospital
**C**	**C**ontext of the situation
**L**	**L**ife-**L**imiting comorbidities
**S**	“**S**igns of life” during conventional CPR (ROSC, motor response during CPR)

## Data Availability

The original contributions presented in the study are included in the article, further inquiries can be directed to the corresponding author.

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
