# Peer review of "CALL TO ECLS—Acronym for Reporting Patients for Extracorporeal Cardiopulmonary Resuscitation Procedure from Prehospital Setting to Destination Centers"

_healthcare, 2024, doi:10.3390/healthcare12161613_

Round 1

Reviewer 1 Report

Comments and Suggestions for Authors

 This paper with the name: Acronym CALL TO ECLS are estimated and tried to show/might prove helpful for Emergency Medical Teams. The objective of this study is to assess the CALL TO ECLS acronym and validate its content to ensure that the acronym encompasses an appropriate number of elements and adequately covers the scope of the studied field. They used a  questionnaire survey using the LimeSurvey platform (to me not fully understood. Delphi method.

The study was conducted over a 6-month period (from December 2022 to May 2023). 

As a result the preliminary version of the tool was sent to 287 experts and only 35 of the answered the questionary.

It is only 10% of all possible cases.

This findings/reports or material can therefore not be estimated or reviewd by me.  

Author Response

Dear Reviewer,

Thank you for your comments regarding our article "Acronym CALL TO ECLS." We would like to address the issues you raised and clarify certain aspects of our study.

The use of LimeSurvey in our study allowed us to distribute the questionnaire and collect data in accordance with the requirements of our University. Although it may not be the most widely known platform, it is well-recognized in the scientific community for its capabilities in survey research. The Delphi method is a well-established research method used to obtain expert opinions and reach consensus in various fields. It involves multiple rounds of questionnaires to gather opinions from a group of experts with the aim of reaching a consensus.

Response Rate

We are aware that a response rate of 10% (35 responses out of 287 invitations sent) is low. We would like to highlight a few points:

  • Engaging experts in research can be challenging, especially considering their busy schedules and other professional commitments.
  • Considering the ECPR program is still being implemented in many countries, many issues remain unclear. Often, an entire team works on the program, and only the leader responds on behalf of the whole team or institution.

Although the response rate was only 10%, we believe that the responses obtained from qualified experts still provide valuable insights. The results may not be fully representative of the entire expert population; however, they provide a basis for further research and analysis. Like any acronym, it requires validation over a period of time, which we intend to do.

We would like to thank you once again for your comments and assure you that we will take them into account in the further development of our project. We look forward to continued discussion and collaboration to improve the quality and usefulness of the CALL TO ECLS acronym for emergency medical teams.

Reviewer 2 Report

Comments and Suggestions for Authors

Dear Authors,

hope my suggestions will improve quality of your paper. The suggestions are mainly in accordance to make your paper clear and concise.

My suggestions are as follow:

The abstract indicates the methodology and the paper's aims. Still, it doesn't have a more concise explanation how the results would affect actual life application and effectiveness of the ECPR acronym in practice - the abstract should give a much more straightforward explanation about how the research improves the ability to implement ECPR. It would be helpful to state specific benefits from the research. 

The introduction identifies the interconnection between the importance of effective communication and the research goals. How communication plays a role in emergency medical outcomes is a relevant topic and subject area being discussed. However, this would be well-served by having more of the closely linked thematics come to bear upon how the research would be conducted around the validation of the CALL TO ECLS. It would be good to establish a much more closely related link between the need for this acronym and the potential positive effects it could engender in remedying the issues presented in the literature pertaining to communication breakdowns and their consequences. You can explicitly state how the acronym could address certain communication barriers identified in the literature.

The methods section is well-articulated with regard to the details of the survey process; however, it could be improved to describe how each survey question directly applies to the evaluation of the acronym. This would give the readers a much more precise notion of how the survey results were directly relevant and supportive of the validity and effectiveness of the acronym.

In the discussion is essential to indicate what future research is needed to explore the effectiveness of these interventions, possibly in different emergency settings or concerning other populations. This should not only conclude the current findings but also open avenues for subsequent inquiries and practical application.

The counclusion is short and concise but it lack the linkage with the overall objective. Yes, it does serve as communication facilitator but the general goal you set is broader in the context of patient security.

Hope this suggestions were helpful to you, good luck and kind regards

Author Response

Dear Reviewer,

Thank you for your valuable comments and suggestions regarding our article. We appreciate the time you have taken to provide them and the constructive criticism that will help us improve the quality of our work. Below are our responses to your individual suggestions:

Abstract

We agree that the abstract should more specifically explain how the results of our study will impact the practical application and effectiveness of the CALL TO ECLS acronym. We will include more direct explanations regarding the benefits arising from the research, such as improving the ability to implement ECPR and the specific advantages resulting from our findings.

Introduction

Thank you for the suggestion to create a clearer link between effective communication and the objectives of the study. In line with your suggestion, we will add further information on how the CALL TO ECLS acronym can address the communication problems identified in the literature.

Methodology

We will supplement the methodology section with this information so that readers have a clear understanding of the relationship between the survey results and the validity and effectiveness of the acronym.

Discussion

We will incorporate your suggestion to indicate future research directions.

Once again, thank you for your suggestions. We are confident that the changes we implement will enhance the clarity and value of our article.

Sincerely,

Autors

Reviewer 3 Report

Comments and Suggestions for Authors

Dear Authors,

It was one of the best articles I've read recently. Survival rates with functional treatments for both in-hospital and out-of-hospital cardiac arrest are extremely low. Extracorporeal cardiopulmonary resuscitation (ECPR) is emerging as a modality to improve prognosis by increasing vital end organ perfusion with extracorporeal membrane oxygenation (ECMO) during conventional CPR and stabilize the patient for interventions to reverse the etiology of arrest. Performing this emergency procedure requires a significant investment in resources, and even the most successful ECPR programs can burden healthcare systems, clinicians, patients, and their families with unrescuable patients supported by extracorporeal devices. Non-randomized and observational studies have repeatedly compared ECPR to conventional CPR after in-hospital cardiac arrest in selected patient populations and have repeatedly demonstrated an association between improved survival. Recently, randomized controlled trials suggest a benefit of ECPR over standard resuscitation. It has been reported that such studies may also be beneficial in out-of-hospital cardiac arrest in healthcare systems with a high level of coordination. Application of these data to clinical practice should be done with caution. Results are likely to vary depending on the environment and system in which ECPR is initiated. Significant ethical challenges are raised, including whether ECPR should be considered an extension of CPR, at what point it becomes extended organ support therapy, and how to approach patients who cannot recover or receive a heart transplant. The economic impact of ECPR varies by healthcare system and has the potential to exceed resources if used indiscriminately. Ideally, studies should include economic evaluations to inform healthcare systems about the cost-benefits of this treatment.

 Suggestion: References should be arranged in Arabic numerals, not Roman numerals.

Author Response

Dear Reviewer, Thank you for your positive comments regarding our article and for the valuable suggestions that will help us improve it.
We agree that the survival rates of patients following sudden cardiac arrest, both in-hospital and out-of-hospital, are extremely low. ECPR indeed emerges as a promising method to improve prognosis by increasing organ perfusion with ECMO during conventional cardiopulmonary resuscitation and stabilising the patient to perform interventions that reverse the cause of the cardiac arrest.
We fully agree that implementing the ECPR procedure requires significant resources and may pose a burden on healthcare systems, clinicians, patients, and their families. These considerations are crucial from both ethical and economic perspectives.
We also concur that the latest randomised studies suggest the superiority of ECPR over standard resuscitation, and their results may be useful even in cases of out-of-hospital cardiac arrest in highly coordinated healthcare systems. However, as you rightly pointed out, the application of these data in clinical practice should be done cautiously, considering the diversity of environments and systems in which ECPR is initiated.
Thank you for the suggestion regarding the organisation of references. We will adhere to it and change the numbering of references to Arabic numerals to ensure consistency and readability of the article.

Round 2

Reviewer 1 Report

Comments and Suggestions for Authors

This paper can not be published in current form due to low/mini respons rate

Author Response

Thank you for the valuable feedback on our manuscript. We understand that the response rate is an important factor in determining the reliability and generalizability of the study results.

We are aware that a 10% response rate (35 responses out of 287 invitations sent) is low. We would like to highlight a few points:

Engaging experts in research can be challenging, especially considering their tight schedules and other professional commitments. Given that the ECPR program is still being implemented in many countries, many aspects remain unclear. Often, the entire team works on the program, but only the leader responds on behalf of the whole team or institution.

Although the response rate was only 10%, we believe that the responses obtained from qualified experts still provide valuable insights. The results may not fully represent the entire population of experts; however, they serve as a basis for further research and analysis. Like any acronym, it requires validation over time, which we intend to undertake.

We are also considering conducting a follow-up study with a larger sample size after a year and communicating a call to action (CALL TO ECLS) to emergency medical services teams.